# Inhibition of SARS-CoV-2 Replication by a Small Interfering RNA Targeting the Leader Sequence

**DOI:** 10.3390/v13102030

**Published:** 2021-10-08

**Authors:** Beatrice Tolksdorf, Chuanxiong Nie, Daniela Niemeyer, Viola Röhrs, Johanna Berg, Daniel Lauster, Julia M. Adler, Rainer Haag, Jakob Trimpert, Benedikt Kaufer, Christian Drosten, Jens Kurreck

**Affiliations:** 1Applied Biochemistry, Institute of Biotechnology, Technische Universität Berlin, 10623 Berlin, Germany; tolksdorf@tu-berlin.de (B.T.); viola.roehrs@tu-berlin.de (V.R.); johanna.berg@tu-berlin.de (J.B.); 2Institute of Chemistry and Biochemistry, Freie Universität Berlin, 14195 Berlin, Germany; chuanxnie@zedat.fu-berlin.de (C.N.); daniel.lauster@fu-berlin.de (D.L.); haag@zedat.fu-berlin.de (R.H.); 3German Centre for Infection Research (DZIF), Charitéplatz 1, 10117 Berlin, Germany; daniela.niemeyer@charite.de (D.N.); christian.drosten@charite.de (C.D.); 4Institute of Virology, Charité-Universitätsmedizin Berlin, 10117 Berlin, Germany; 5Department of Veterinary Medicine, Institute of Virology, Freie Universität Berlin, 14163 Berlin, Germany; j.adler@fu-berlin.de (J.M.A.); jakob.trimpert@fu-berlin.de (J.T.); b.kaufer@fu-berlin.de (B.K.); 6Department of Infectious Diseases and Respiratory Medicine, Charité-Universitätsmedizin Berlin, 10117 Berlin, Germany

**Keywords:** SARS-CoV-2, COVID-19, RNAi therapy, siRNA, 5′-UTR, leader sequence

## Abstract

Severe acute respiratory syndrome coronavirus 2 (SARS-CoV-2) has infected almost 200 million people worldwide and led to approximately 4 million deaths as of August 2021. Despite successful vaccine development, treatment options are limited. A promising strategy to specifically target viral infections is to suppress viral replication through RNA interference (RNAi). Hence, we designed eight small interfering RNAs (siRNAs) targeting the highly conserved 5′-untranslated region (5′-UTR) of SARS-CoV-2. The most promising candidate identified in initial reporter assays, termed siCoV6, targets the leader sequence of the virus, which is present in the genomic as well as in all subgenomic RNAs. In assays with infectious SARS-CoV-2, it reduced replication by two orders of magnitude and prevented the development of a cytopathic effect. Moreover, it retained its activity against the SARS-CoV-2 alpha variant and has perfect homology against all sequences of the delta variant that were analyzed by bioinformatic means. Interestingly, the siRNA was even highly active in virus replication assays with the SARS-CoV-1 family member. This work thus identified a very potent siRNA with a broad activity against various SARS-CoV viruses that represents a promising candidate for the development of new treatment options.

## 1. Introduction

Coronavirus disease 2019 (COVID-19) is an infectious disease caused by the severe acute respiratory syndrome coronavirus 2 (SARS-CoV-2). The virus was first described in Wuhan, People’s Republic of China, in December 2019 and spread worldwide very quickly. As of August 2021, approximately 200 million infections and more than 4 million fatalities have been reported worldwide. Despite the remarkable development and approval of vaccines, satisfactory treatment options for patients with a severe course of disease are still lacking. So far, only remdesivir, a ribonucleotide analogue inhibitor of viral RNA polymerase that was originally developed to treat Ebola infections but is also active against SARS-CoV-2 [1], has received conditional approval by the European medicines agency (EMA) [2]. Furthermore, neutralizing monoclonal antibodies have been identified that block SARS-CoV-2 [3] and are therefore a class of biologics that is recommended for the treatment. Still, the development of effective antiviral drugs is urgently needed.

SARS-CoV-2 is an enveloped RNA virus with a high genomic homology to SARS-CoV-1 (82%) and belongs to β Coronavirus (Figure 1A). The single-stranded RNA genome is of positive polarity, approximately 30 kilobases in length and codes for 16 non-structural proteins (nsp1–16) that are responsible for RNA replication, as well as for 4 structural proteins, namely the spike (S) protein, the envelope (E) protein, the membrane (M) protein and the nucleoprotein (N). The S, E and M proteins are embedded in the virus membrane that envelops the nucleocapsid, which is composed of the N protein and the virus genome [4,5]. The S protein is responsible for binding and entering the host cell and consists of two subunits [6]. The S1 subunit contains the receptor binding domain (RBD), which conveys the binding to the host cell receptor angiotensin-converting enzyme 2 (ACE2). After cleavage, e.g., by the cellular protease transmembrane protease serine 2 (TMPRSS2), cell entry is subsequently mediated by the S2 subunit [7].

Following cell entry, all proteins required for replication are directly translated (ORF1ab) and induce replication of the viral RNA genome (Figure 1A). During replication, negative-sense RNA intermediates are produced to act as templates for positive-sense RNA synthesis [8]. A unique feature of coronaviruses and other members of the order *Nidovirales* is that not only the complete genomic RNA, but also subgenomic RNAs, are produced (Figure 1A), which are essential for the translation of the structural proteins. These subgenomic RNAs all share the same 5′-end (and 3′-end) of around 75 nucleotides, the so-called leader sequence [9]. The current model of discontinuous transcription proposes that the viral RNA-dependent RNA polymerase (RdRp) switches template from the transcription regulatory sequences (TRS) that precede all of the ORFs (body-TRS) to the TRS in the 5′-untranslated region (5′-UTR) (leader-TRS) during synthesis of the negative strand. This ultimately results in the fusion of the leader sequence to downstream parts of the genome. Subsequently, these negative-sense strand intermediates are transcribed into positive-sense strand subgenomic mRNAs [10]. As soon as sufficient new RNA genomes and structural proteins are available, new viruses are formed and released [11].

A promising approach to inhibit viral replication is the implementation of RNA interference (RNAi), which is a highly conserved, very efficient process of post-transcriptional gene silencing that is triggered by short double-stranded RNA, such as small interfering RNA (siRNA). RNAi leads to the degradation of mRNA in a sequence-specific manner and ultimately specifically inhibits the expression of genes [12,13,14]. Numerous RNAi-based therapies have been investigated for the treatment of viral infections in the past [15,16]. We and others have demonstrated an effective reduction of target viral gene expression and virus replication, for example of the human immunodeficiency virus [17], influenza virus [18], coxsackievirus B3 [19,20], adenovirus [21], hepatitis B virus [22] and hepatitis C virus [23] in various cell and animal disease models. Following the emergence of SARS-CoV-1 in 2002, RNAi was one of the strategies pursued to inhibit SARS-CoV-1 replication. In several in vitro studies, it was shown that siRNAs efficiently reduced the expression of different target viral proteins and ultimately led to the inhibition of viral replication [24,25]. Owing to its fundamental role in cell infection, the S protein of SARS-CoV-1 was considered the most promising target for inhibition with RNAi. However, the S protein is prone to mutations, which increase the virus fitness, as has also been observed for SARS-CoV-2 in a growing number of SARS-CoV-2 variants (e.g., alpha (B.1.1.7), beta (B.1.351), gamma (P.1) and delta (B.1.617.2) variants) [26]. Obviously, mutations can alter the target sequence of siRNAs and severely reduce the efficiency of RNAi [27,28,29]. Therefore, targeting a strongly conserved region such as the 5′-UTR that is crucial for viral RNA replication and transcription was considered a promising target [30,31]. By targeting the leader sequence of SARS-CoV-1, which is present in the viral genomic as well as subgenomic RNAs, Li et al. were able to effectively inhibit the expression of viral genes (S, E, M and N) and ultimately virus replication in Vero E6 cells [32]. Moreover, the siRNA they developed targeting the leader sequence was more effective than the siRNA they developed in parallel targeting the S gene. Hence, the 5′-UTR of SARS-CoV-2 is another promising target to develop RNAi-based therapies against COVID-19.

In the present study, we designed and tested eight siRNAs targeting the 5′-UTR of SARS-CoV-2. The most efficient siRNA, termed siCoV6, targets the leader sequence that is present in the viral genome as well as in all subgenomic fragments. When tested with infectious SARS-CoV-2, it effectively inhibited virus replication by two orders of magnitude (99%) at only 10 nM siRNA. As its target sequence is highly conserved, it was also active against the SARS-CoV-2 alpha variant and even against a SARS-CoV-1 family member. This broad antiviral activity makes the siRNA a promising candidate for the treatment of SARS- CoV-2 infections.

## 2. Materials and Methods

### 2.1. The Design of siRNA and Plasmid Construction

Eight siRNA duplexes targeting the 5′-UTR of SARS-CoV-2 (Figure 1B) were designed in silico using Horizon siDesign. The absence of seed sequences in the human transcriptome was ensured by the Nucleotide BLAST program using default parameters. The control siRNA siCon does not match any sequence present in the viral or human genome [33]. All siRNAs were purchased from Eurofins Genomics (Ebersberg, Germany) and their sequences are shown in Table 1.

The silencing efficacy of the siRNAs was determined in reporter assays using a vector expressing the firefly luciferase reporter gene fused with the 5′-UTR of SARS-CoV-2. The 5′-UTR of SARS-CoV-2 (NCBI RefSeq NC_045512.2) was synthesized by Thermo Fisher Scientific (Waltham, MA, USA) and inserted downstream of the *Renilla*-luciferase gene into the psiCHECK™ 2 Vector (Promega, Fitchburg, WI, USA) plasmid using the restriction sites XhoI/NotI, giving rise to psiCheck2 SARS-CoV-2 5′-UTR.

### 2.2. Cell Culture

Vero E6 cells (CRL-1586, ATCC, Manassas, VA, USA) were cultivated in Dulbecco’s modified eagle’s medium (DMEM, Biowest, Nuaillé, France), 10% fetal calf serum (FCS, c.c.pro, Oberdorla, Germany), 1% sodium pyruvate (Sigma, Steinheim, Germany), 1% non-essential amino acids (NEAA, Biowest) and 1% penicillin/streptomycin (Biowest). For propagation of Hela cells (ACC 57, DSMZ, Braunschweig, Germany), DMEM was supplemented with 10% FCS, 1% NEAA, 1% penicillin/streptomycin and 1% L-Glutamin (Biowest). Both cell lines were incubated at 37 °C and 5% CO2 in a humidified atmosphere.

### 2.3. Dual-Luciferase Reporter Assay

Hela cells (10^5^) were seeded in 24-well plates, and 250 ng of the reporter plasmid psiCheck2 SARS-CoV-2 5′-UTR was co-transfected with 40 nM siRNA 24 h later using Lipofectamin 2000 (Thermo Fisher Scientific, Waltham, MA, USA) according to the manufacturer’s instructions. Silencing efficacy of the siRNAs was determined after 48 h by measuring relative luciferase activity using the Dual-Luciferase Reporter Assay System (Promega, Fitchburg, WI, USA) according to the manufacturer’s protocol.

### 2.4. Viral Replication Assay

All SARS-CoV-related infection experiments were performed under biosafety level-3 (BSL-3) conditions with enhanced respiratory personal protection equipment at Charité-Universitätsmedizin Berlin and Freie Universität Berlin. Vero E6 cells (3 × 10^5^ cells/mL) were transfected with 0.1–150 nM siCoV6 or 100 nM siCon 24 h after seeding using Lipofectamin RNAiMAX (Thermo Fisher Scientific, Waltham, MA, USA) according to the manufacturer’s protocol. Cells were washed with PBS 24 h after transfection and infected with SARS-CoV-2 (BetaCoV/Munich/BavPat2-ChVir984-ChVir1017/2020), SARS-CoV-2 alpha variant (BetaCoV/Germany/ChVir21652/2020) or SARS-CoV-1 (recombinant SARS-CoV-1, GenBank: AY310120.1) [34], respectively, at a MOI of 0.01. Viral RNA was extracted from 50 μL of culture supernatant at the indicated time points using viral RNA isolation kits (Macherey-Nagel, Analytik, Jena, Germany) according to the manufacturer’s protocol. Genome equivalents per mL (GE/mL) were determined by quantitative RT-PCR using probes for the E gene as previously reported [35,36].

### 2.5. Cytopathogenicity

Cells were fixed in 4% formaldehyde (Roth, Karlsruhe, Germany) 48 h after infection for 24 h and the CPE of viral infection was documented with a Zeiss Observer Z1 microscope (Zeiss, Jena, Germany).

### 2.6. Statistical Analyses

Statistical significance was analyzed by one-way ANOVA or Student’s *t*-test using GraphPad Prism 8 software (La Jolla, CA, USA). All data were collected in triplicate in at least three independent experiments and are represented as mean ± standard error of the mean (SEM). Statistical significance is shown as * *p*  ≤  0.05, ** *p*  ≤  0.01, *** *p*  ≤  0.001 or **** *p*  ≤  0.0001.

## 3. Results

### 3.1. Design of siRNAs Targeting the 5′-UTR of SARS-CoV-2

The 5′-UTR of SARS-CoV-2 is a conserved region that is crucial for viral RNA replication and transcription, which makes it a promising target for the development of RNAi-based therapies against COVID-19 [30,31]. As the efficiency of siRNAs strongly depends on their thermodynamic design and the structure of the target region [37], eight siRNAs targeting the 5′-UTR of SARS-CoV-2 (NCBI RefSeq NC_045512.2) were designed in silico (Figure 1B, Table 1) using the online tool Horizon siDesign. Three of these siRNAs (siCoV3, siCoV6, siCoV7) bind to highly conserved regions that are completely homologous to SARS-CoV-1 (Figure 1B). The absence of seed sequences in the human transcriptome was ensured by the Nucleotide BLAST program using default parameters. The control siRNA siCon does not match any sequence present in the viral or human genome [33].

### 3.2. RNAi-Mediated Targeting of the 5′-UTR of SARS-CoV-2 in Reporter Assays

The silencing activity of the designed siRNAs was determined in a dual-luciferase reporter assay. For the assay, the reporter plasmid psiCheck2 SARS-CoV-2 5′-UTR (Figure 2A) expressing the *Renilla*-luciferase reporter gene and fused with the 5′-UTR of SARS-CoV-2 was used. Hela cells were transiently co-transfected with the reporter plasmid and one of the eight different siRNA candidates to evaluate their silencing activity. The relative Ren/Luc activity was determined 48 h after transfection. The expression of *Renilla*-luciferase was significantly inhibited by all eight siRNAs, with siCoV6, siCoV2, siCoV5 and siCoV4 achieving inhibition rates of around 90% (Figure 2B). These results demonstrate that the 5′-UTR of SARS-CoV-2 can be efficiently targeted by a number of different siRNAs. We chose the most effective candidate, siCoV6, which binds to the TRS of the leader sequence, for our subsequent tests.

### 3.3. RNAi-Mediated Inhibition of SARS-CoV-2 Replication

After having shown that the siRNA is active against the SARS-CoV-2 5′-UTR in reporter assays, we tested siCoV6 against the infectious SARS-CoV-2 virions in Vero E6 cells. This siRNA is particularly promising, as it targets the TRS in the leader sequence and is thus directed against the genomic and subgenomic RNA transcripts. It can therefore be expected to be particularly efficient, as the RNAi approach should degrade not only the genomic RNA, but also all subgenomic mRNAs. For the virus inhibition assays, Vero E6 cells were transfected with varying concentrations of siCoV6 (0.1 nM to 150 nM) and infected with SARS-CoV-2 at a multiplicity of infection (MOI) of 0.01 after 24 h. Viral RNA was isolated from the supernatant 24 h and 48 h after infection and genome equivalents per mL (GE/mL) were determined by quantitative RT-PCR [36]. The tested concentrations of siCoV6 led to a significant inhibition of the replication of SARS-CoV-2 after 24 h in a concentration-dependent manner (Figure 3A).

In the control infection, 7.91 × 10^8^ GE/mL were detected, while already the lowest tested concentration of 0.1 nM siCoV6 resulted in a reduction of approximately one order of magnitude to 9.37 × 10^7^ GE/mL. The most effective inhibition was seen at the highest concentration of 150 nM siCoV6, with a reduction to less than 1% of the control value to 7.64 × 10^6^ GE/mL. The extent of virus inhibition decreased at 48 h post infection. The detected genomic equivalents were still reduced by up to 97%, for example to 8.36 × 10^7^ GE/mL with 100 nM siCoV6 compared to 2.85 × 10^9^ GE/mL measured for the control. However, the reduction no longer reached statistical significance.

The protective effect of the siRNA treatment was further investigated by documenting the cytopathic effect (CPE) by transmitted light microscopy. As expected, no CPE was visible in mock infected control cells (Figure 3B). In clear contrast, infection with SARS-CoV-2 in untreated cultures (no siRNA) and cultures treated with the control siRNAs (siCon) induced visible CPE characterized by rounding of the cells and detachment from the cell culture plate. In line with the observed RNAi-mediated inhibition of SARS-CoV-2 with siCoV6, siRNA treatment strongly reduced the CPE. In particular, a concentration of 50 nM siCoV6 or more almost completely inhibited the onset of CPE. These results clearly demonstrate that siCoV6 inhibits replication of SARS-CoV-2 in a concentration-dependent manner and protects cells from infection-induced cell toxicity.

### 3.4. Inhibition of the SARS-CoV-2 Alpha Variant by siCoV6

Like other viruses, SARS-CoV-2 is constantly evolving, as seen by a growing number of SARS-CoV-2 variants [26]. Mutations can result in the alteration of siRNA target sequences and severely reduce the efficiency of RNAi [27,28,29]. Regions of the viral genome that are highly conserved are less likely to accumulate mutations that are not detrimental to virus replication and are preferred target sites for siRNAs. Therefore, three of the siRNAs (siCoV3, siCoV6, siCoV7) were designed to target highly conserved regions of the coronavirus genome (Figure 1B). One of the siRNAs, siCoV6, was specifically designed against the highly conserved TRS in the leader sequence. It can therefore be expected to be not only very efficient as it targets the genomic RNA as well as all subgenomic transcripts, but also to be active against SARS-CoV-2 variants.

To test this hypothesis, we next examined whether siCoV6 can also inhibit the replication of the SARS-CoV-2 alpha variant. To this end, Vero E6 cells were transfected with 100 nM of siCoV6 and genome equivalents per mL (GE/mL) were determined as described above [35,36]. As can be seen in Figure 4A, siRNA treatment of the cells inhibited replication of the SARS-CoV-2 alpha variant significantly by approximately 7-fold from 1.78 × 10^8^ GE/mL in the control culture to 2.54 × 10^7^ GE/mL in the treated culture 24 h post infection. As already observed for SARS-CoV-2, the inhibitory activity became less prominent at 48 h post infection. The virus load was reduced by 70% from 5.02 × 10^8^ GE/mL in the cells treated with the control siRNA to 1.61 × 10^8^ GE/mL in cells treated with the active siCoV6.

We also documented the CPE 48 h post infection by transmitted light microscopy (Figure 4B). As expected, no CPE became visible in the mock infected control cells, while infection with the SARS-CoV-2 alpha variant induced a strong CPE in untreated (no siRNA) and siCon-treated cells. In sharp contrast, as already observed for SARS-CoV-2, siCoV6 strongly reduced the occurrence of a CPE by the infection of SARS-CoV-2 alpha variant.

Currently, other SARS-CoV-2 variants, and in particular the delta variant, have become predominant in many countries worldwide. We therefore carried out a bioinformatic analysis of approximately 1500 sequences of different SARS-CoV-2 variants (NCBI Virus SARS-CoV-2 Data Hub, as of June 2021) with respect to mutations in the target site of siCoV6. In this analysis, we did not identify any mutations in either the SARS-CoV-2 beta or delta variant. For the alpha and gamma variants, only less than 0.8% of the isolates had a mutation in the target site of siCoV6, but the important seed region of siCoV6 was never affected. This confirms that the targeted TRS is particularly well suited for the development of a broad-range RNAi therapeutic.

### 3.5. Inhibition of SARS-CoV-1 by siCoV6

As siCoV6 targets a sequence that is conserved between SARS-CoV-1 and SARS-CoV-2 (Figure 1B), we reasoned that it may also be active against the first SARS-coronavirus. We therefore tested the inhibitory efficiency of siCoV6 against SARS-CoV-1 under the same conditions as in our previous experiments. As can be seen in Figure 4A, RNAi treatment inhibited virus replication by approximately two orders of magnitude to 1.06 × 10^9^ GE/mL compared to the control in which 1.24 × 10^11^ GE/mL were determined after 24 h. Again, the effect substantially decreased after 48 h and no longer reached significant levels.

We furthermore investigated the CPE 48 h post infection by transmitted light microscopy (Figure 4C). As expected, mock infected control cells showed no sign of CPE, while infection with SARS-CoV-1 resulted in the cytopathic phenotype observed before in untreated (no siRNA) and siCon-treated cultures. In contrast, as already observed for SARS-CoV-2 and the SARS-CoV-2 alpha variant, siCoV6 strongly reduced the occurrence of a CPE.

Taken together, we demonstrated that an siRNA designed to target the TRS in the 5′-UTR of SARS-coronaviruses is highly active against SARS-CoV-2, including its variant alpha. Importantly, a substantial inhibitory effect was also observed for the more distantly related SARS-CoV-1. Our siRNA might therefore have the potential to be developed into an anti-coronavirus therapy with a broad target range.

## 4. Discussion

The novel human coronavirus disease, COVID-19, caused by SARS-CoV-2 has developed into a devastating pandemic. Ongoing research aims to identify therapeutic strategies for the prevention and treatment of COVID-19. However, despite the approval of vaccines against SARS-CoV-2 and increasing immunization rate of the population, satisfactory treatment options for patients with severe disease courses are still lacking [38]. Targeting gene expression by RNAi has become a standard method in biomedical research and the technology is being developed as a new therapeutic strategy with a growing number of siRNAs being approved by the authorities. It is also an auspicious approach to inhibit viral replication as frequently investigated in the past [15,16]. In particular, RNA viruses, like SARS-CoV-2, are promising candidates, as not only their mRNAs (as in case of DNA viruses), but also their genomic RNA can be directly targeted by RNAi. In accordance with this, we and others have shown in the past that the replication of RNA viruses, like the human immunodeficiency virus [17], influenza virus [18], coxsackievirus B3 [19,20] and hepatitis C virus [23], was effectively inhibited by RNAi in various cell and animal disease models.

In the present study, we designed siRNAs directed against the 5′-UTR of the SARS-CoV-2 genome by selecting a conserved region that is crucial for viral RNA replication and transcription, which previous experience has shown is a promising target [30,31,39]. As the efficiency of siRNAs depends on numerous factors such as their thermodynamic design and the structure of the target site [37] and their efficiency cannot be predicted, we designed a set of eight siRNAs. Reporter assays confirmed that all eight designed siRNAs efficiently targeted the 5′-UTR of SARS-CoV-2 (Figure 1B) and led to a significant downregulation of the luciferase reporter activity (Figure 2B). The most efficient siRNA out of this panel of tested siRNAs, named siCoV6, binds to the highly conserved TRS and therefore targets the leader sequence. The leader sequence, consisting of around 75 nucleotides at the 5′-end of the SARS-CoV-2 genome, is also incorporated into the subgenomic RNAs by a process of discontinuous transcription. This unusual mechanism makes the leader sequence an attractive target, as siRNAs will not only destroy the genomic RNA of SARS-CoV-2, but also the subgenomic RNAs (Figure 1A), which are essential for the translation of the structural proteins and hence the replication of the virus [9]. In fact, siCoV6 effectively inhibited replication of SARS-CoV-2 and protected cells from a CPE (Figure 3). Moreover, as its target sequence is highly conserved, siCoV6 also showed promising inhibitory activity against the SARS-CoV-2 alpha variant and even against SARS-CoV-1 (Figure 4).

A challenge for the development of siRNA therapeutics with antiviral activity is that viruses mutate rather quickly, which frequently allows them to alter the target sequence of the siRNAs and severely reduce or completely abolish the efficiency of RNAi [27,28,29]. After the outbreak of SARS-CoV-1, several in vitro studies investigated siRNAs directed against the S protein of SARS-CoV-1 due to its fundamental role in cell infection [24,40,41,42,43]. Moreover, the combination of two siRNAs targeting the S and N proteins also potently suppressed SARS-CoV-1-induced pathogenesis in vivo, as shown in a Rhesus macaque SARS-CoV model [44]. However, the S and N proteins are prone to mutations that do not reduce viral fitness, which is also observed in SARS-CoV-2 by an increasing number of SARS-CoV-2 variants [26]. To prevent escape mutants, it is therefore desirable to target a highly conserved region like the leader sequence, as demonstrated in a study by Li et al. in which an siRNA directed against the leader sequence of SARS-CoV-1 significantly inhibited virus replication in Vero E6 cells [32]. They showed that the siRNA inhibited the expression of all the structural viral genes (S, E, M and N) and was more effective than an siRNA targeting the S gene alone. The target site of this siRNA directed against the leader sequence of SARS-CoV-1 is, however, mutated in the SARS-CoV-2 genome at two positions including one in the seed region, which can be expected to render the siRNA inefficient. In contrast, the siRNA siCoV6 designed in the present study targets a highly conserved sequence that is unaltered in the SARS-CoV-2 alpha variant and SARS-CoV-1 genomes.

A slight reduction in the efficiency of siCoV6 against the SARS-CoV-2 alpha variant (Figure 4A) can be explained by possible differences in the local target structure rather than mutations in the target site [45,46,47]. When analyzing approximately 1500 of the most recent complete sequences of the different SARS-CoV-2 variants (NCBI Virus SARS-CoV-2 Data Hub), we did not find any mutations in the siCoV6 target site for the SARS-CoV-2 delta variant and the SARS-CoV-2 beta variant. For the SARS-CoV-2 alpha variant and the SARS-CoV-2 gamma variant, only <0.8% of sequences showed mutations in the siCoV6 target site. The seed region of siCoV6 was, however, never affected. As we have experimentally shown for the alpha variant as an example that siCoV6 maintains its activity, this bioinformatic analysis confirms the eligibility of the 5′-UTR and in particular the leader sequence as a target site for the implementation of RNAi as a broad-range antiviral approach [31,39].

Several other in silico studies predicted siRNAs targeting ORF1ab [48,49,50,51], S gene [49,51,52,53], ORF3 [49,51], E gene [49], M gene [49,51] and N gene [51,53] of SARS-CoV-2. Two in vitro studies further demonstrated a robust downregulation of S, M and N on the mRNA or protein levels by siRNAs in overexpression cell culture models [54,55]. In a recent in vivo study by Idris et al., siRNAs targeting the RdRp and helicase of SARS-CoV-2 were successfully delivered by lipid nanoparticle (LNP) formulations to the lung of a K18-hACE2 mouse model by intravenous injection. The siRNA-LNPs robustly repressed virus in the lungs and provided a survival advantage to the treated mice [56]. Another recent in vivo study by Khaitov et al. demonstrated that topical application by inhalation of a stabilized siRNA modified with a peptide dendrimer targeting the RdRp of SARS-CoV-2 significantly reduced viral load and inflammation in the lung of a Syrian hamster model [57]. Taken together, these are promising results for the development of RNAi-based therapy of COVID-19. However, as previously mentioned, targets like the S Protein or RdRp have the drawback that they are prone to mutations [58,59]. The target site of the most promising siRNA designed in the present study is the leader sequence that is highly conserved and enables the simultaneous targeting of the genomic RNA and all subgenomic RNAs. In analogy to combination therapy with low molecular weight drugs, potential viral escape can be suppressed with a combination of siRNAs [60,61,62], for example, of siCoV6 with siCoV7, which also binds in a highly conserved region (Figure 1B); though, given the very high level of conservation of the leader sequence in SARS-CoV-2 and SARS-CoV-1, the viral escape potential is probably very low.

In this study, commercial Lipofectamin RNAiMAX was used for siRNA transfection, which is not recommended for in vivo use. The lung is the major site of a COVID-19-infection due to high expression of ACE2, and long-term lung damage can be a severe complication of COVID-19 [63]. In the past few decades, substantial progress has been made regarding the improvement of siRNA stability and efficiency of delivery to the respiratory system [12,15,64,65,66]. For example, nasally applied siRNAs resulted in the highly effective inhibition of respiratory syncytial virus (RSV) and SARS-CoV-1 in animal models [44,67,68]. Nebulizers or inhalers were developed to generate aerosols for drug delivery by inhalation directly to the lung [69] as recently successfully implemented for SARS-CoV-2 inhibition [57]. Administration of an siRNA targeting the RSV N gene (ALN-RSV01) by inhalation for the potential treatment or prevention of RSV infection was also investigated in a clinical study. The siRNA and application route were well tolerated and may have beneficial effects and prevent *bronchiolitis obliterans* syndrome in lung transplant patients infected with RSV [70,71,72]. We therefore envision the delivery of siCoV6 to the lung by inhalation as a plausible administration route for the treatment of COVID-19. The next steps will be to test the siRNAs described here and the proposed route of delivery in our advanced bioprinted 3D lung model consisting of a base of primary lung fibroblasts and macrophage-like cells that were overlayed with an alveolar epithelial cell line [73] and subsequently in a COVID-19 hamster model [74].

A special feature of coronaviruses among RNA viruses is the existence of a proofreading mechanism, which hampers the development of nucleoside analogs as therapeutics as they are removed during replication and transcription [75]. At the same time, the proofreading activity is favorable for RNA-targeting approaches as it increases genomic stability, thereby reducing the risk of the development of escape mutants. In our study, we demonstrated the high antiviral activity of siCoV6 against SARS-CoV-2. Its target site is widely conserved among the currently available SARS-CoV-2 variants, and we experimentally demonstrated that the siRNA maintains its activity against SARS-CoV-2 alpha and even against SARS-CoV-1. This broad antiviral activity makes siCoV6 a promising candidate for the treatment of SARS coronavirus infections.

## 5. Patents

The Technische Universität Berlin, Freie Universität Berlin and Charité, Universitätsmedizin have filed a patent application for the siRNAs described in the present study with B.T., C.N., D.N., V.R., J.B., D.L., J.T., B.K. and J.K. as authors.

## Figures and Tables

**Figure 1 viruses-13-02030-f001:**
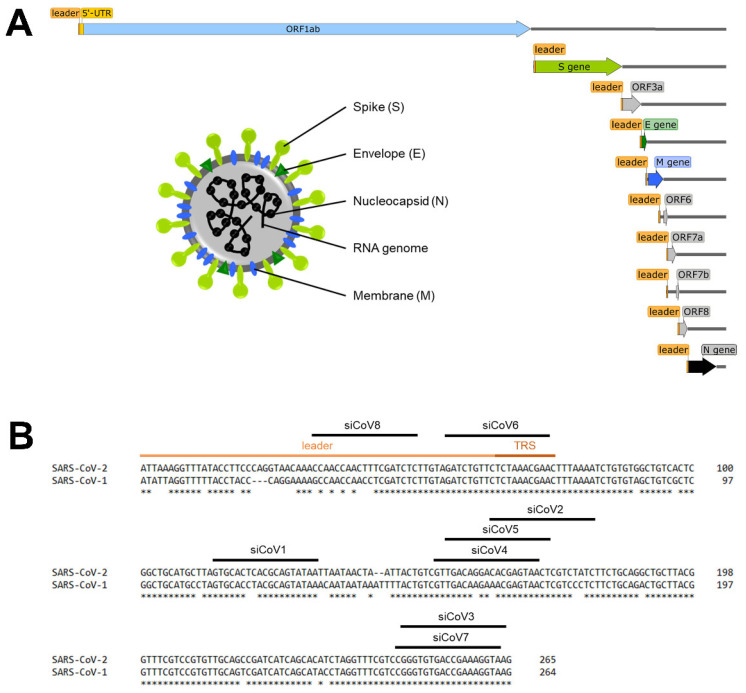
Schematic overview of the SARS-CoV-2 genome and target sequences of the siRNAs in the 5′-UTR. (**A**) Schematic overview of the SARS-CoV-2 genome, the subgenomic RNAs and the virion structure. The coronavirus virion consists of the structural spike protein (S), envelope protein (E), membrane protein (M) and nucleoprotein (N). The single-stranded RNA genome with positive polarity is encapsulated by N, while S-trimers protrude from the host-derived virus envelope and enable binding to new host cells. In addition to the genomic RNA, nine subgenomic RNAs are synthesized during replication, all of which share the same 5′- and 3′-ends. The orange box represents the mutual 5′-end, the leader sequence. (**B**) Target sequences of the siRNAs in the 5′-UTR of SARS-CoV-2. The leader sequence (light orange) and the transcription regulatory sequence (TRS, dark orange) can be found at the 5′-end of the genomic and subgenomic RNAs. Nucleotides without an asterisk differ from the SARS-CoV-1 5′-UTR sequence.

**Figure 2 viruses-13-02030-f002:**
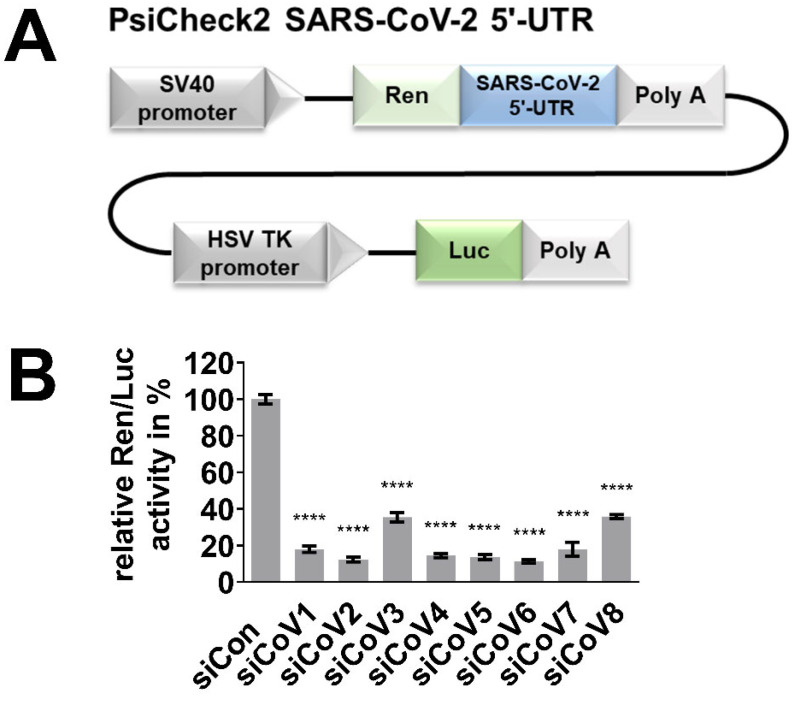
Efficient targeting of the 5′-UTR of SARS-CoV-2 by the designed siRNAs. The silencing activity of the designed siRNAs was examined in a dual-luciferase reporter assay. (**A**) Schematic representation of the reporter plasmid psiCheck2 SARS-CoV-2 5′-UTR. (**B**) Hela cells were co-transfected with 40 nM of the respective siRNA and 250 ng of the reporter plasmid. Proteins were extracted 48 h after transfection, the relative Ren/Luc activity was determined and the negative control (siCon) was set to 100%. The mean ± SEM of three independent experiments is shown. The statistical significance was determined by a univariate analysis of variance (one-way ANOVA); **** *p* < 0.0001.

**Figure 3 viruses-13-02030-f003:**
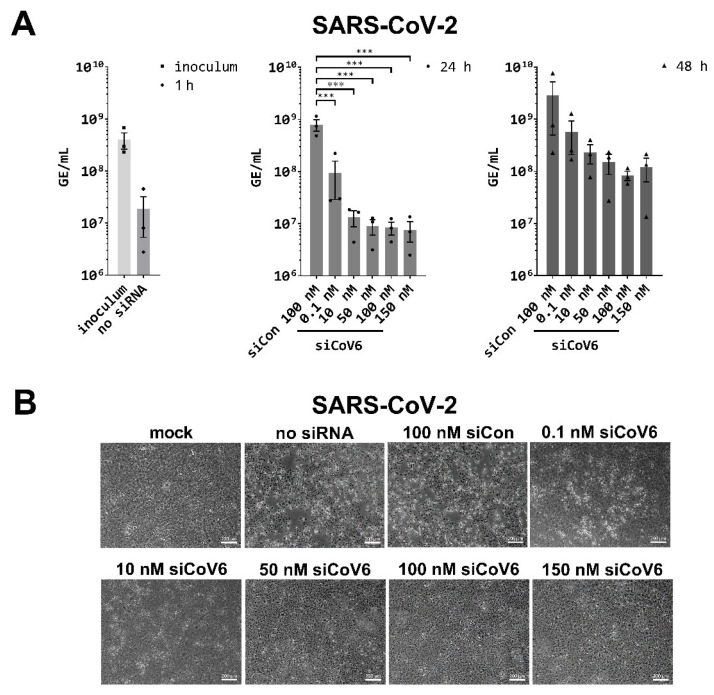
Inhibition of SARS-CoV-2 replication by siCoV6. (**A**) The replication of SARS-CoV-2 (BetaCoV/Munich/BavPat2-ChVir984-ChVir1017/2020) was investigated by quantitative RT-PCR in Vero E6 cells. Vero E6 cells were transfected with 0.1–150 nM siCoV6 or 100 nM siCon and infected 24 h after transfection with SARS-CoV-2 at a MOI of 0.01. Subsequently, the cells were washed, supplemented media were added and viral RNA was isolated from the culture supernatant at 1 hpi, 24 hpi and 48 hpi. Genome equivalents per mL (GE/mL) were determined by quantitative RT-PCR. The mean ± SEM of three independent experiments is shown. The statistical significance was determined by a univariate analysis of variance (one-way ANOVA); *** *p* < 0.001. (**B**) Cell morphology observation. Cells were fixed 48 h after infection with SARS-CoV-2 and the cytopathic effect (CPE) of viral infection was documented by transmitted light microscopy. Representative data from three independent experiments are shown. Scale bar: 200 µm.

**Figure 4 viruses-13-02030-f004:**
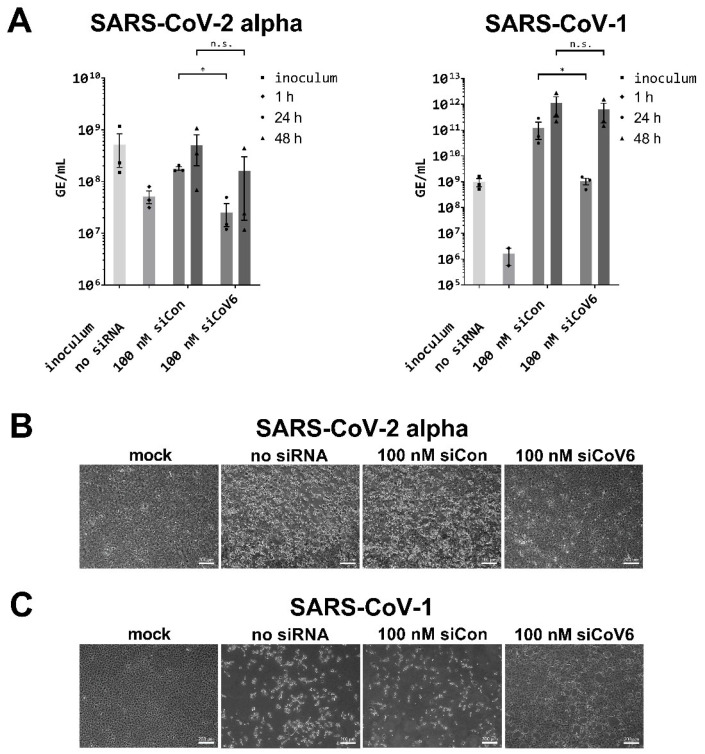
Inhibition of SARS-CoV-2 alpha and SARS-CoV-1 replication by siCoV6. (**A**) The replication of SARS-CoV-2 alpha (BetaCoV/Germany/ChVir21652/2020) and SARS-CoV-1 (recombinant SARS-CoV-1, GenBank: AY310120.1) was investigated by quantitative RT-PCR in Vero E6 cells. Vero E6 cells were transfected with 100 nM siCoV6 or 100 nM siCon and infected 24 h after transfection with SARS-CoV-2 alpha or SARS-CoV-1 at a MOI of 0.01. Genome equivalents per mL (GE/mL) were determined by quantitative RT-PCR as described in Figure 3. The mean ± SEM of three independent experiments is shown. The statistical significance was determined by Student’s *t*-test; * *p* < 0.05. (**B**,**C**) Cell morphology observation. Cells were fixed 48 h after infection with SARS-CoV-2 alpha or SARS-CoV-1 and the cytopathic effect (CPE) of viral infection was documented by transmitted light microscopy. Representative data from three independent experiments are shown. Scale bar: 200 µm.

**Table 1 viruses-13-02030-t001:** Sequences of designed siRNAs.

siRNA	Sequence
siCoV1	5′-UUAUACUGCGUGAGUGCACTT-3′ 3′-TTAAUAUGACGCACUCACGUG-5′
siCoV2	5′-GAUAGACGAGUUACUCGUGTT-3′ 3′-TTCUAUCUGCUCAAUGAGCAC-5′
siCoV3	5′-UUACCUUUCGGUCACACCCTT-3′ 3′-TTAAUGGAAAGCCAGUGUGGG-5′
siCoV4	5′-UUACUCGUGUCCUGUCAACTT-3′ 3′-TTAAUGAGCACAGGACAGUUG-5′
siCoV5	5′-AGUUACUCGUGUCCUGUCATT-3′ 3′-TTUCAAUGAGCACAGGACAGU-5′
siCoV6	5′-UUCGUUUAGAGAACAGAUCTT-3′ 3′-TTAAGCAAAUCUCUUGUCUAG-5′
siCoV7	5′-UACCUUUCGGUCACACCCGTT-3′ 3′-TTAUGGAAAGCCAGUGUGGGC-5′
siCoV8	5′-GAGAUCGAAAGUUGGUUGGTT-3′ 3′-TTCUCUAGCUUUCAACCAACC-5′
siCon	5′-ACGUGACACGUUCGGAGAATT-3′ 3′-TTUGCACUGUGCAAGCCUCUU-5′

## Data Availability

Not applicable.

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
