# Peer review of "Inhibition of SARS-CoV-2 Replication by a Small Interfering RNA Targeting the Leader Sequence"

_viruses, 2021, doi:10.3390/v13102030_

Round 1
Reviewer 1 Report
This is a very interesting and very well-written study. Only a minor comment. Since in your title you mention small interefering RNAs but only actually present one, I woud like to either see the inhibition efficacies of the other 7 siRNAs or modify your title (do not use plural).
Author Response
Dear Reviewer 1,
Thank you for your very positive evaluation of our manuscript.
We followed your suggestion and modified our title. We now use the singular:
Inhibition of SARS-CoV-2 replication by a small interfering RNA targeting the leader sequence
Thank you and best regards
Jens Kurreck
Reviewer 2 Report
The study of Tolksdorf et al. reports the use of siRNA targeting SARS-CoV-2 replication. They identified a siRNA with a high activity on sars-Cov-2 replication using différent strains including sars-cov-1, sars-cov-2.
Can the authors specify the gene investigated during qRTPCR on supernatant.
The quality of the pictures in all figure should be improved.
In the figure 3 which sars-cov-2 strain was used ?
Author Response
Dear Reviewer 2,
Thank you for your very positive evaluation of our manuscript. We made the following changes according to your suggestions:
- Can the authors specify the gene investigated during qRTPCR on supernatant.
Thank you for reminding us of adding this important information. On line 170/171, we added that we amplified the viral region of the E gene. - The quality of the pictures in all figure should be improved.
Thank you, we improved the quality of the figures. We added versions of higher resolution (1200 dpi). - In the figure 3 which sars-cov-2 strain was used ?
In Figure 3, the SARS-CoV-2 strain BetaCoV/Munich/BavPat2-ChVir984-ChVir1017/2020 was used. This information was added to the figure legend. To be consistent throughout the manuscript, we added in the legend to Figure 4 that we used the following alpha variant: BetaCoV/Germany/ChVir21652/2020 and the following SARS-CoV-1 strain: recombinant SARS-CoV-1, GenBank: AY310120.1
Best regards
Jens Kurreck